# TG/HDL Ratio Is an Independent Predictor for Estimating Resting Energy Expenditure in Adults with Normal Weight, Overweight, and Obesity

**DOI:** 10.3390/nu14235106

**Published:** 2022-12-01

**Authors:** Annaliese Widmer, Margaret G. Mercante, Heidi J. Silver

**Affiliations:** 1Department of Medicine, Vanderbilt University Medical Center, Nashville, TN 37232, USA; 2College of Arts and Sciences, Vanderbilt University, Nashville, TN 37212, USA; 3Department of Veterans Affairs, Tennessee Valley Healthcare System, Nashville, TN 37212, USA

**Keywords:** insulin resistance, obesity, resting energy expenditure, TG/HDL ratio

## Abstract

Factors that determine resting energy expenditure (REE) remain under investigation, particularly in persons with a high body mass index (BMI). The accurate estimation of energy expenditure is essential for conducting comprehensive nutrition assessments, planning menus and meals, prescribing weight and chronic disease interventions, and the prevention of malnutrition. This study aimed to: (a) determine the contribution of cardiometabolic biomarkers to the inter-individual variation in REE in persons categorized by BMI; and (b) assess the contribution of these biomarkers in the prediction of REE when persons of varying BMI status were categorized by their glycemic and metabolic syndrome status. Baseline data from 645 adults enrolled in diet intervention trials included REE measured by indirect calorimetry, body composition by dual energy X-ray absorptiometry, anthropometrics, and cardiometabolic biomarkers. Multivariate linear regression modeling was conducted to determine the most parsimonious model that significantly predicted REE by BMI category, metabolic syndrome status, and glycemic status. Modeling with the traditional predictors (age, sex, height, weight) accounted for 58–63% of the inter-individual variance in REE. When including age, sex, height, weight and fat-free mass as covariates, adding TG/HDL to regression modeling accounted for 71–87% of the variance in REE. The finding that TG/HDL is an independent predictor in estimating REE was further confirmed when participants were categorized by metabolic syndrome status and by glycemic status. The clinical utility of calculating the TG/HDL ratio not only aids health care providers in identifying patients with impaired lipid metabolism but can optimize the estimation of REE to better meet therapeutic goals for weight and disease management.

## 1. Introduction

The accurate estimation of daily energy expenditure is essential for conducting comprehensive nutrition assessments, planning menus and meals, prescribing weight and chronic disease interventions, preventing malnutrition, and determining parenteral and enteral nutrition support formulations. Factors that determine energy expenditure in humans remain under investigation, particularly in persons with a high body mass. Resting energy expenditure (REE) continues to be the most frequently measured constituent of total daily energy expenditure (TEE), as it comprises ~60–70% of TEE (with the thermic effect of food comprising ~10% and physical activity energy expenditure comprising the remaining 15–30%). Prior studies show that the prediction of REE including age, sex, height, weight, fat and/or fat-free mass as independent variables accounts for ~65% of the inter-individual variability in REE [1,2]. As current evidence does not support a robust genetic effect, with heritability estimates of REE being only 0.3 MJ/day [3], it is likely that a significant portion of the unexplained inter-individual variance in REE is related to cardiometabolic risk factors including biomarkers indicative of impaired glucose or lipid metabolism and insulin resistance [3].

The findings of studies in adults with obesity on the relationship between glycemia and REE are ambiguous. Whereas one study in adults with obesity and type 2 diabetes detected no correlation between fasting plasma glucose or HbA1c and REE [4], another showed that including fasting plasma glucose in regression modeling improved the prediction of REE by 3% [5]. Further, glucose disposal and fasting insulin level were significant determinants of REE in Pima peoples, independent of age, sex and body composition [6]. The relationship between glucose, insulin, insulin resistance and REE has also been observed in an inter-generational study of 149 families [3]. More recently, a positive association between insulin resistance and REE has been shown in normal and overweight healthy adults after adjustment for age and sex [7]. As insulin resistance is an early pathological indicator for the development of T2D, the higher REE observed in persons with T2D [3,8] and the metabolic syndrome [9] may be a function of their insulin resistance.

However, the gold standard method of determining the presence of insulin resistance, the hyperinsulinemic-euglycemic clamp [10], is a costly, time-consuming and invasive approach used primarily in research settings. Consequently, surrogate markers of insulin resistance are more widely used, including the HOMA-IR and HOMA-2IR scores. Yet, the measurement of blood insulin level is not a routinely performed component of standard clinical practice, assay methods have not been standardized [11], and it is an expensive test. Thus, other non-insulin-based surrogate indicators of insulin resistance that are more widely available across health care settings include the calculation of the ratio of triglyceride to HDL-cholesterol (TG/HDL) and the triglyceride glucose index (TyG). Both an increased plasma triglyceride level and a decreased HDL-cholesterol level are independent predictors of insulin resistance [12]. An increased TG/HDL ratio has been observed in persons with impaired fasting glucose, prediabetes, T2D and the metabolic syndrome compared to normoglycemic individuals [13,14,15]. Moreover, analysis from 10,132 participants of the U.S. National Health and Nutrition Examination Survey (NHANES) showed that each 0.1 unit increase in the TG/HDL ratio was associated with a 51% increased risk of insulin resistance after adjustment for demographic, anthropometric and clinical covariates [16].

Beyond being a surrogate biomarker of insulin resistance, an elevated TG/HDL ratio may be an indicator of imbalance in the delivery and uptake of lipids to the liver, and consequently, dysfunctional hepatic lipid metabolism that contributes to the pathophysiology of insulin resistance. Indeed, the TG/HDL ratio is independently associated with non-alcoholic fatty liver disease [17]. Moreover, it is now under investigation that changes in the liver, including the increased delivery of free fatty acids to the liver in the state of high adiposity (obesity), when subcutaneous adipose tissue has reached its limit of expansion to store excess lipid, alters endocrine and paracrine functions to cause insulin resistance [18]. Notably, more recent metabolomic and lipidomic methods have identified lipid families and subclasses that function in regulating insulin sensitivity and action [19]. Yet, scientific debate continues as to the causative role and specific mechanisms underlying the impact of lipid classes on the development of insulin resistance.

The investigation of the combined effects of the traditional predictors (sex, age, height, weight) and clinically available cardiometabolic biomarkers on REE to better approximate daily energy (caloric) needs in persons of varying body mass status is limited. Therefore, the purpose of the current study was to: (1) determine the cardiometabolic biomarkers that significantly contribute to the inter-individual variation in REE after adjustment for age, sex, height, weight and fat or fat-free mass in individuals with normal weight, overweight, and obesity; and (2) assess the contribution of cardiometabolic biomarkers in the prediction of REE when individuals with a normal weight, overweight and obesity are categorized by their glycemic and metabolic syndrome status.

## 2. Methods

### 2.1. Subjects

The present findings derive from compiling baseline data from adults who participated in one of five diet intervention trials conducted at the Vanderbilt Diet, Body Composition and Human Metabolism Core between 2011 and 2021. Subjects were recruited by email distribution lists announcing the trials and enrolled if they met the following eligibility criteria: age ≥ 18 years, BMI ≥ 18.5 kg/m^2^, weight stable for at least 3 months prior to enrollment, no diagnosis of cancer, heart, liver, lung, kidney or thyroid disease, no diagnosis of infectious or auto-immune disease, no prior bariatric surgery, not taking medications affecting appetite or weight, no food allergies or dietary restrictions and not pregnant or lactating. Approvals were obtained from the Ethics Committee of the Vanderbilt University Medical Center Institutional Review Board, and all subjects provided written informed consent prior to study visits. All physical and metabolic assessments were conducted at the Vanderbilt Clinical Research Center, with subjects arriving in a 10 h overnight fasted state. Upon arrival, vital signs and a fasting venous sample were obtained by a clinical research center nurse.

### 2.2. Clinical Labs

Whole blood samples were submitted to the Vanderbilt Department of Pathology Diagnostic Laboratory for analysis. Serum glucose was measured by the colorimetric timed endpoint method, and serum insulin was measured by chemiluminescent immunoassay. Serum triglycerides, total cholesterol and HDL-cholesterol were assayed by selective enzymatic hydrolysis. The homeostatic model assessment of insulin resistance (HOMA-IR) was calculated as ([fasting glucose (mM) × fasting insulin (mU/L)]/22.5) and HOMA-%β using the online HOMA2 calculator [20,21]. Metabolic syndrome status was classified according to the 2009 joint statement criteria [22]. Glycemic status was classified based on ICD-code diagnosis in the electronic medical record and fasting plasma glucose level: normoglycemic (no diagnosis and glucose < 100 mg/dL), prediabetes (diagnosis or glucose 100–125 mg/dL), diabetes (diagnosis or glucose ≥ 126 mg/ dL) [23].

### 2.3. Anthropometry and Body Composition

Height (±0.1 cm), weight (±0.1 kg) and waist and hip circumferences (±0.1 cm) were measured in triplicate by a research dietitian using standardized procedures. Whole and regional body composition were measured by one certified densitometrist using a Lunar iDXA™ (GE Healthcare, Madison, WI, USA), which was phantom-calibrated each morning before data collection to ensure instrument reliability. Coefficients of variation (CV) from repeated measures performed in 12 randomly selected subjects were <1.5% for total fat, total lean, trunk fat and trunk lean masses, ensuring the precision and reliability of the DXA measurements [24]. Visceral adipose tissue (VAT) mass was quantified using the CoreScan algorithm in Encore software version 13.6 (GE Healthcare, Madison, WI, USA), which computes VAT mass (g) in the android region.

### 2.4. Indirect Calorimetry

Resting energy expenditure (REE) was measured in the overnight fasted state under standard thermoneutral conditions using a metabolic cart system (ParvoMedics TrueOne 2400^®^, Sandy, UT, USA). Before each study visit, the system was calibrated to room air and a single gas tank (~16% O_2_, 1% CO_2_). The whole-body rates of oxygen consumption (O_2_) and carbon dioxide (CO_2_) production were determined from measurements of expired volume and the differences in O_2_ and CO_2_ between inspired and expired air. Ventilation was measured by a mass flow meter, oxygen concentration was measured by a paramagnetic O_2_ analyzer and carbon dioxide was measured by an infrared CO_2_ analyzer. Data were collected for a total of 15 min upon subjects having rested in the supine position for ≥10 min and reaching a steady state under the hood with an average change in minute VO_2_ ≤ 10% and an RQ ≤ 5% [25,26]. No subjects had an RQ < 0.7 or >1.0. Measurement of VO_2_ and VCO_2_ in liters per minute enabled the automated calculation of REE via the Weir equation [27]. Substrate oxidation rates as a percentage of REE, after adjustment for 24 h urinary urea nitrogen excretion, were automatically calculated using the methods of Frayn [28].

### 2.5. Statistical Analysis

Continuous variables were checked for normality by the visual inspection of histograms and the Shapiro–Wilk test. Descriptive statistics are summarized for continuous variables as the mean ± standard deviation and for categorical variables as the frequency and percentage. Preliminary analysis showed no significant differences between participants with Class I (BMI 30.0–34.9 kg/m^2^) and Class II (BMI 35.0–39.9 kg/m^2^) obesity for age, HOMA-IR, TG/HDL ratio, REE or the variance accounted for in REE with the traditional model (age, sex, height, weight). Thus, final analyses were conducted with subjects assigned to one of four BMI categories: normal weight (18.5–24.9 kg/m^2^), overweight (25.0–29.9 kg/m^2^), Class I/II obesity (30.0–39.9 kg/m^2^) and Class III obesity (≥40 kg/m^2^). Univariate associations between independent demographic, clinical, and body composition variables and REE were explored by Pearson’s correlation coefficients and simple linear regression to identify potential predictors for inclusion in multivariate regression modeling. A forward stepwise approach was used for multivariate linear regression modeling to determine the most parsimonious model that significantly predicted REE by BMI category and again with subjects categorized by metabolic syndrome and glycemic status. Statistical analysis was performed using SPSS version 28.0 (IBM, Montauk, NY, USA).

## 3. Results

Of the 645 participants who met the eligibility criteria (Figure 1), 380 (58.9%) were female, 265 (41.1%) were male, 418 (64.8%) self-identified as white and 227 (35.2%) self-identified as Black, with no significant differences observed in the proportion of participants in each of the four BMI categories by sex or race/ethnicity. Overall, 38 (5.9%) participants were normal weight, 86 (13.3%) were overweight, 442 (68.6%) had class I/II obesity and 79 (12.2%) had class III obesity (Table 1). The average age was 42.2 ± 13.5 years, with obese participants being significantly older than normal weight participants by 7–10 years (F = 6.413, *p* < 0.001).

The primary outcome, resting energy expenditure, was significantly lower in normal weight participants compared to all other BMI groups (all *P*s < 0.001), with a mean difference of an additional 266.9 ± 67.2 kcal/day being expended over 24 h in overweight participants, 247.9 ± 58.4 kcal/day in obesity class I/II participants and 492.8 ± 69.4 kcal/day in obesity class III participants. The REE in class III participants was also significantly higher than that in overweight and class I/II participants by a mean difference of 225–245 kcal/day (*P*s < 0.001). There was no difference in respiratory quotient across the four groups (*p* = 0.24), which averaged 0.82 ± 0.06, and no significant differences among groups were observed in substrate utilization, as the proportions of fat, carbohydrates and protein being oxidized during metabolic testing were similar (all *P*s > 0.20).

### 3.1. Differences in Linear Regression Models by BMI Category

Univariate analysis confirmed that there were significant associations between the traditional predictors of age, sex, height, weight and REE (Table 2). In addition, REE correlated with fat-free mass (*r* = 0.76, *p* < 0.001), HOMA-IR score (*r* = 0.19, *p* < 0.001) and TG/HDL ratio (*r* = 0.34, *p* < 0.001). The univariate relationship between REE and TG/HDL ratio was more robust than the relationship between REE and HOMA-IR score (adjusted *R*^2^ = 0.11 vs. 0.03, Figure 2). Measurements of regional and total fat (amounts and percentages), including VAT, were not significantly associated with REE.

Multivariate regression modeling with the traditional predictors (age, sex, height, weight) accounted for 58–63% of the variance in REE: normal weight adjusted *R*^2^ = 0.62, overweight adjusted *R*^2^ = 0.63, obese class I/II adjusted *R*^2^ = 0.58, obese class III adjusted *R*^2^ = 0.61 (Table 3, Model 1). Adjusting the traditional model for fat-free mass increased the amount of the variability in REE, accounting for only 1% in the normal weight group but for 16% in the overweight group and for 7% in the Class I/II and Class III obese groups (Table 3, Model 2).

Within these groups, there were significant differences in TG/HDL ratios and HOMA-IR scores. The TG/HDL ratio was significantly higher in overweight, obese Class I/II and obese Class III participants compared to normal weight participants (1.68 ± 0.99, 2.63 ± 2.03, 3.26 ± 2.05, respectively, vs. 1.12 ± 0.55, all *P*s < 0.001). Adding the TG/HDL ratio to the models further increased the amount of the variance in REE accounted for by 16% in the normal weight group, 8% in the overweight group and 6% in the obese Class I/II and Class III groups (Table 3, Model 3). Thus, 71–87% of the overall inter-individual variability in REE was accounted for in the models that included the TG/HDL ratio.

The HOMA-IR score was 4.2–5.5 units higher in obesity Class III participants compared to normal weight, overweight and obesity Class I/II participants (all *P*s < 0.001). However, adding the HOMA-IR score to the models decreased the amount of the variance in REE accounted for in the normal weight and overweight groups and only increased the variance accounted for by 2% in the Class I/II obesity group and by 1% in the Class III obesity group (Table 3, Model 4).

### 3.2. Differences in Linear Regression Models by Glycemic and Metabolic Syndrome Status

To further assess the impact of the TG/HDL ratio as a predictor of REE, we conducted two sub-analyses; first, participants were grouped by their glycemic status and then by their metabolic syndrome status. A comparison of descriptive characteristics by subgroups is presented in Table 4 and Table 5. Among the 645 participants, the REE was 162.5 ± 41.4 kcal/day higher in those with type 2 diabetes (T2D) compared to normoglycemic participants and 174.7 ± 47.2 kcal/day higher compared to those with prediabetes (*P*s < 0.001), but no difference was detected in REE between the normoglycemic and prediabetic participants (12.2 ± 34.4 kcal/day, *p* = 0.93).

As with REE, there was no significant difference detected in HOMA-IR scores between normoglycemic and prediabetic participants (*p* = 0.48). However, the TG/HDL ratio differed among the three groups from 2.0 ± 1.5 in normoglycemic participants to 2.7 ± 1.8 in prediabetic participants and 4.3 ± 2.7 in those with T2D (all *P*s < 0.001). As shown in Table 6 (Models 2 and 3), adding fat-free mass to the traditional model (age, sex, height, weight) and including the TG/HDL ratio as a predictor variable increased the amount of the inter-individual variance accounted for in predicting REE by 11–13%. In contrast, adding HOMA-IR to these models (Table 6, Model 4) either reduced or increased the variance accounted for in predicting REE by only 1%.

The REE, HOMA-IR score and TG/HDL ratio were also significantly higher when comparing participants by metabolic syndrome status (Table 5), with 125.6 ± 29.4 kcal more being expended daily by participants with metabolic syndrome (*p* < 0.001) compared to those without metabolic syndrome. Both the HOMA-IR score and TG/HDL ratio were >2 times higher in those with metabolic syndrome. Consistent with the findings when participants were grouped by glycemic status, including the TG/HDL ratio in the final model increased the inter-individual variability accounted for in predicting REE by 9–12% when participants were grouped by metabolic syndrome status (Table 7).

## 4. Discussion

The major novel finding of the present study, conducted in a large cohort of adults with a wide range of age and BMI, is that TG/HDL ratio is an independent predictor of REE. Importantly, the TG/HDL ratio was significantly associated with the REE in both univariate and multivariate regression analyses. It is noteworthy that the estimation of REE in these 645 participants, who ranged in age from 18 to 81 years, showed substantial improvement when adjusting for TG/HDL ratio. Overall, 71–87% of the inter-individual variability in estimating REE was accounted for when the traditional predictors (age, sex, height, weight) were adjusted for fat-free mass and TG/HDL ratio.

An elevated TG/HDL ratio has long been associated with having an atherogenic lipid profile and, more recently, with being in a state of insulin resistance [29,30]. Thus, the TG/HDL ratio is a biomarker for both impaired glucose and lipid metabolism. Indeed, a high TG/HDL ratio is associated with insulin resistance in children, adolescents, and adults of various ages and BMIs [16,31,32,33,34], and much prior evidence demonstrates that the TG/HDL ratio is equal or more accurate when compared to any other surrogate biomarker of insulin resistance. It is interesting that both the normoglycemic participants and participants who did not meet the criteria for having metabolic syndrome encompassed a range of BMIs and of hyperinsulinemia. While some studies have suggested that the TG/HDL ratio is only a biomarker for insulin resistance in Caucasian or white persons, data from larger longitudinal or population-based studies show a similar relationship in Hispanics, African Americans, and Asians, as observed with Caucasians [35,36,37]. In the present study, we found no effect on the inter-individual variability in REE when including race/ethnicity in regression models.

The steeper slope of the regression line that was observed for the association between TG/HDL ratio and REE compared to the association between HOMA-IR and REE indicates not only a stronger relationship between TG/HDL ratio and REE but that an increase in the TG/HDL ratio is associated with a greater increase in REE. Supporting these findings, when participants were categorized by glycemic or metabolic syndrome status, we found that the TG/HDL ratio was a robust independent predictor of REE. Further, adding HOMA-IR to the regression models did not improve the proportion of the variance accounted for in predicting REE.

Beyond the current awareness of TG/HDL ratio as a biomarker of insulin resistance, it has become evident that an elevated TG/HDL ratio may be indicative of the lipid handling dysfunction in the liver and skeletal muscle that likely causes insulin resistance. In the normal weight individual, it is expected that metabolically healthy subcutaneous adipose tissue (SAT) will store large amounts of energy in the form of triglycerides. In the obese state, it is more likely that the dysregulation of SAT expansion and storage will occur, and lipids will begin to accumulate in other depots. Excess lipid deposits in the intraabdominal space (visceral adiposity) are part of a complex pathophysiological phenotype that includes the release of free fatty acids and the storage of triglycerides in the liver, in skeletal muscle, adjacent to and surrounding the heart, in the pancreas, and in the kidneys. This accumulation of ectopic fat disrupts organ and tissue function, promoting a state of impaired sensitivity or response to insulin action, i.e., insulin resistance. Intrahepatic triglyceride content is a robust predictor of insulin action, not only in the liver but also in skeletal muscle and adipose tissue [38]. In a normal (non-hyperinsulinemic) state, the oxidation of VLDL-TG would contribute 10–20% to resting energy expenditure [39]. However, a state of chronic hyperinsulinemia is associated with overproduction and reduced clearance of VLDL-triglyceride. While de novo lipogenesis typically supplies about 5% to the hepatic triglyceride pool, the investigation of non-alcoholic fatty liver disease has shown that this contribution increases to 25–40% in a state of hyperglycemia and hyperinsulinemia. Such dysregulation of lipid metabolism may be a result of diet (i.e., high saturated fat or high simple sugar dietary intake) and/or genetics [40]. Of relevance is that these factors likely induce mitochondrial dysfunction and oxidative stress from the overproduction of reactive oxygen species [41]. Evidence from rodent and in vitro models has shown that hepatic steatosis is a marker for this impaired metabolic milieu [42].

Another distinctive finding from the present study was that REE did not differ between persons with Class I and Class II obesity but was significantly higher in those with Class III obesity (BMI ≥ 40 kg/m^2^). Limited evidence has been published on REE in persons with severe obesity. The present data suggest that the impact of having a high body mass on REE becomes most influential when both body mass and insulin resistance are severely high. Baseline data from bariatric surgery patients with Class III obesity showed that those with insulin resistance had a higher REE than those who were insulin-sensitive, even when adjusted for fat-free mass [43]. Both the HOMA-IR score and TG/HDL ratio were significantly higher in participants with Class III obesity compared to those with Class I/II obesity. In Class III obese participants, the univariate and multivariate relationship between TG/HDL ratio and REE was stronger than that between HOMA-IR score and REE, which is in line with the other findings of this study.

It is interesting that we did not detect any difference in substrate oxidation or respiratory quotient among the BMI groups. While it is theorized that increased fatty acid oxidation would produce increased energy expenditure, little published evidence supports the concept that switching to greater fat oxidation increases energy expenditure. However, the flow of the substrate to storage or oxidation is controlled by insulin action. To date, there remains several mechanisms under study to explain ectopic fat, other factors, and pathways associated with reduced insulin action. One area of continued investigation and debate is the potential for altered mitochondrial activity and function in the state of insulin resistance, which may affect bioenergetics. The evidence remains conflicting, as some studies show a compensatory increase in mitochondrial oxidative capacity in insulin resistance, while others show decreased or unchanged capacity [44,45,46].

The limitations of the present study include that we did not assess all conceivable factors that may associate with REE, such as cardiorespiratory fitness or physical activity level. Adding time spent in moderate-to-vigorous physical activity had a small effect in estimating REE in young adults [47]. It is also possible that the energy cost of exercise and weight-bearing activities, including movement, differs among persons with varying BMI status. Secondly, cross-sectional data do not allow for elucidating the factors involved in the dynamic nature of energy expenditure. Lastly, the measurement of body composition by more technologically advanced imaging methods, such as CT and MRI, may be superior to DXA, although DXA is more widely available and cost-effective. Moreover, our prior data showed only slight differences in measurements of whole body and regional fat between MRI and DXA, with a small potential overestimation of lean tissue by DXA [24]. The strengths of the present study include the large sample size, the presence of a wide range of ages, BMI status, and glycemic and insulinemic statuses among participants and the use of the gold standard indirect calorimetry method for measuring REE, which was performed in the fasted state under well-controlled thermoneutral environmental conditions.

In conclusion, the results indicate that adjusting the classical covariates associated with REE (age, sex, height, weight) for fat-free mass (representing metabolically active organs and tissue) substantially increases the variance accounted for in the prediction modeling of persons who are overweight and have Class I, II or III obesity. However, there remains a significant portion of the inter-individual variability in REE that can be accounted for by including the TG/HDL ratio. Furthermore, an elevated TG/HDL ratio has proven to be a robust risk factor for impaired fasting glucose, prediabetes, type 2 diabetes, metabolic syndrome and cardiovascular disease in multiple studies. Thus, the utility of calculating the TG/HDL ratio, a parameter widely accessible in standard clinical practice, is not only to aid healthcare providers in identifying patients with lipid handling dysfunction and/or insulin resistance but to optimize the estimation of REE to better determine energy requirements and meet therapeutic goals for weight and chronic disease management.

## Figures and Tables

**Figure 1 nutrients-14-05106-f001:**
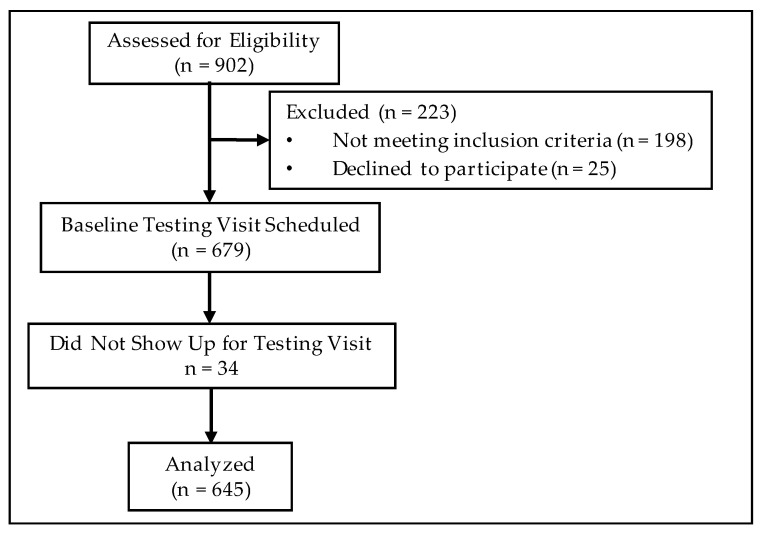
CONSORT Style Flowchart of Subject Recruitment and Retention.

**Figure 2 nutrients-14-05106-f002:**
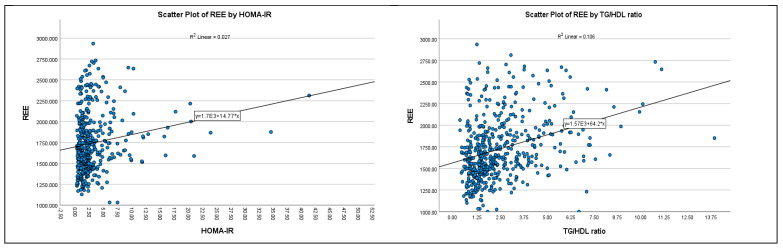
Correlation between HOMA-IR score, TG/HDL ratio and Resting Energy Expenditure (REE) in 645 adults.

**Table 1 nutrients-14-05106-t001:** Characteristics of 645 Study Participants by Body Mass Index (BMI) Category.

	Normal (BMI 18.5–24.9) *n* = 38	Overweight (BMI 25.0–29.9) *n* = 86	Class I/II Obese (BMI 30.0–39.9) *n* = 442	Class III Obese (BMI ≥ 40.0) *n* = 79	*p*-Value
Age (y)	35.5 ± 13.0	39.3 ± 17.9	42.8 ± 12.6	45.4 ± 11.5	<0.001
Weight (kg)	67.3 ± 10.3	85.3 ± 11.9	98.5 ± 14.4	123.0 ± 15.8	<0.001
BMI (kg/m^2^)	23.3 ± 1.5	27.9 ± 1.4	34.6 ± 2.7	43.3 ± 3.4	<0.001
Glucose (mg/dL)	84.2 ± 31.9	93.9 ± 23.7	99.8 ± 28.4	114.7 ± 46.3	<0.001
Insulin (μIU/mL)	6.1 ± 3.6	7.7 ± 8.2	10.9 ± 8.6	18.9 ± 16.5	<0.001
HOMA-IR (score)	1.6 ± 2.3	1.9 ± 2.4	2.9 ± 3.1	7.0 ± 9.3	<0.001
Triglycerides (mg/dL)	66.5 ± 29.1	81.1 ± 35.3	110.4 ± 64.1	129.4 ± 59.2	<0.001
HDL-cholesterol (mg/dL)	62.3 ± 13.6	53.4 ± 13.6	47.5 ± 12.5	44.3 ± 10.8	<0.001
TG/HDL (ratio)	1.1 ± 0.5	1.7 ± 1.0	2.6 ± 2.0	3.3 ± 2.0	<0.001
REE (kcal)	1475.4 ± 369.3	1742.3 ± 365.5	1721.9 ± 338.3	1968.2 ± 362.8	<0.001
% CHO (kcal)	32.1 ± 14.9	32.4 ± 14.7	31.6 ± 15.5	29.3 ± 14.2	0.618
% Fat (kcal)	50.8 ± 18.2	52.8 ± 15.4	49.9 ± 15.7	52.8 ± 14.6	0.276
Metabolic Equivalents (kcal/min)	0.88 ± 0.11	0.82 ± 0.10	0.74 ± 0.11	0.66 ± 0.08	<0.001
Respiratory Quotient (VCO_2_/VO_2_)	0.82 ± 0.07	0.82 ± 0.05	0.82 ± 0.06	0.81 ± 0.05	0.235
Total Tissue Mass (kg) ^a^	77.2 ± 13.6	89.2 ± 12.8	96.9 ± 13.7	119.0 ± 15.4	<0.001
Total Fat Mass (kg)	18.5 ± 9.0	27.1 ± 6.5	40.8 ± 7.6	58.1 ± 9.5	<0.001
Total Fat (% tissue)	28.1 ± 8.4	33.1 ± 8.6	43.8 ± 7.2	49.3 ± 6.2	<0.001
Trunk Fat (kg)	9.5 ± 5.7	15.6 ± 5.0	22.9 ± 5.3	35.4 ± 7.1	<0.001
Trunk Fat (% tissue)	31.5 ± 16.5	37.0 ± 10.4	47.1 ± 7.3	54.5 ± 5.3	<0.001
Visceral Adipose Tissue (kg)	1.1 ± 0.7	1.4 ± 0.9	1.7 ± 1.1	2.5 ± 1.2	<0.001
Android Fat (kg)	1.5 ± 1.1	2.6 ± 1.0	4.0 ± 1.1	6.3 ± 1.4	<0.001
Android Fat (% tissue)	31.8 ± 13.6	40.1 ± 11.2	50.9 ± 7.4	57.0 ± 5.8	<0.001
Gynoid Fat (kg)	3.6 ± 1.5	4.0 ± 1.2	6.6 ± 1.7	9.5 ± 2.1	<0.001
Gynoid Fat (% tissue)	35.9 ± 10.7	33.4 ± 10.5	44.9 ± 9.0	49.8 ± 7.5	<0.001
Total Lean Mass (kg)	45.5 ± 8.4	57.6 ± 14.1	53.2 ± 12.6	59.9 ± 12.2	<0.001
Trunk Lean (kg)	21.9 ± 4.2	27.5 ± 7.1	25.3 ± 5.9	28.8 ± 5.6	<0.001
Android Lean (kg)	2.9 ± 0.7	3.9 ± 0.9	3.8 ± 0.8	4.5 ± 0.9	<0.001
Gynoid Lean (kg)	6.4 ± 1.4	8.9 ± 2.4	8.4 ± 2.1	9.7 ± 2.2	<0.001

^a^ Total tissue mass by DXA = fat mass + lean mass + bone mineral content.

**Table 2 nutrients-14-05106-t002:** Univariate Associations between Predictor Variables and Resting Energy Expenditure in 645 adults.

Variable	rho	95% CI	*p*-Value
Sex (f/m)	0.34	0.26, 0.41	<0.001
Race/Ethnicity (w/b)	−0.17	−0.24, −0.09	<0.001
Age (y)	−0.16	−0.24, −0.08	<0.001
Height (cm)	0.64	0.58, 0.68	<0.001
Weight (kg)	0.66	0.61, 0.71	<0.001
BMI (kg/m^2^)	0.28	0.21, 0.36	<0.001
Glucose (mg/dL)	0.19	0.11, 0.27	<0.001
Insulin (μIU/mL)	0.18	0.09, 0.27	<0.001
HOMA-IR (score)	0.19	0.10, 0.28	<0.001
HOMA-%β	0.05	−0.04, 0.15	0.255
Triglycerides (mg/dL)	0.27	0.16, 0.32	<0.001
HDL-cholesterol (mg/dL)	−0.37	−0.42, −0.27	<0.001
TG/HDL (ratio)	0.34	0.26, 0.42	<0.001
Visceral Adipose Tissue (g)	−0.09	−0.85, 0.79	0.87
Trunk Fat (g)	0.40	−0.07, 0.72	0.08
Android Fat (g)	0.40	−0.07, 0.72	0.08
Gynoid Fat (g)	−0.01	−0.46, 0.45	0.97
Total Fat Mass (g)	0.33	−0.15, 0.68	0.16
Trunk Lean (g)	0.77	0.48, 0.91	<0.001
Android Lean (g)	0.78	0.50, 0.91	<0.001
Gynoid Lean (g)	0.74	0.43, 0.89	<0.001
Total Fat-Free Mass (g)	0.76	0.46, 0.90	<0.001
Total Body Mass (g)	0.82	0.74, 0.88	<0.001
Total Body Fat (%)	0.07	−0.39, 0.51	0.76
Trunk Fat (%)	−0.09	−0.85, 0.79	0.87
Android Fat (%)	0.20	−0.28, 0.60	0.39
Gynoid Fat (%)	−0.26	−0.64, 0.22	0.27

**Table 3 nutrients-14-05106-t003:** Linear Regression Modeling to Predict Resting Energy Expenditure in 645 Normal Weight, Overweight, and Obese Adults.

Variables	Normal Weight BMI				Overweight BMI				Obese Class I/II BMI				Obese Class III BMI			
**Model 1**	**Adjusted *R*^2^ = 0.62, *p* < 0.001**				**Adjusted *R*^2^ = 0.63, *p* < 0.001**				**Adjusted *R*^2^ = 0.58, *p* < 0.001**				**Adjusted *R*^2^ = 0.61, *p* < 0.001**			
	**Estimate**	**St Error**	**t Statistic**	***p* value **	**Estimate**	**St Error**	**t Statistic**	***p* value **	**Estimate**	**St Error**	**t Statistic**	***p* value **	**Estimate**	**St Error**	**t Statistic**	***p* value **
Constant	−930.04	1263.68	−0.736	0.47	−854.30	707.82	−1.207	0.23	−1100.58	239.24	−4.600	<0.001	−1875.3	713.07	−2.630	0.01
Height	7.85	10.73	0.731	0.47	10.45	6.32	1.654	0.10	13.05	2.02	6.454	<0.001	16.72	5.70	2.935	0.005
Weight	14.71	9.74	1.511	0.14	9.71	5.49	1.766	0.08	7.35	1.40	5.248	<0.001	9.92	2.90	3.424	0.001
Sex	246.02	113.41	2.169	0.04	113.92	67.12	1.696	0.09	55.20	25.73	2.146	0.03	−45.46	62.31	−0.730	0.47
Age	−7.12	3.43	−2.075	0.05	−6.21	1.45	−4.296	<0.001	−4.20	0.87	−4.824	<0.001	−2.95	2.31	−1.277	0.21
**Model 2**	**Adjusted *R*^2^ = 0.63, *p* = 0.001**				**Adjusted *R*^2^ = 0.79, *p* < 0.001**				**Adjusted *R*^2^ = 0.65, *p* < 0.001**				**Adjusted *R*^2^ = 0.68, *p* < 0.001**			
	**Estimate**	**St Error**	**t Statistic**	***p* value **	**Estimate**	**St Error**	**t Statistic**	***p* value **	**Estimate**	**St Error**	**t Statistic**	***p* value **	**Estimate**	**St Error**	**t Statistic**	***p* value **
Constant	1894.29	1358.7	1.394	0.19	−1291.30	1087.25	−1.188	0.24	363.55	277.06	1.312	0.19	−38.13	849.88	−0.045	0.96
Height	−14.42	11.51	−1.252	0.23	13.50	9.92	1.362	0.18	2.19	2.27	0.963	0.34	3.21	6.77	0.474	0.64
Weight	10.68	9.73	1.097	0.29	2.92	7.37	0.396	0.69	1.51	1.45	1.044	0.30	5.29	3.12	1.702	0.09
Sex	−60.21	118.61	−0.508	0.62	−10.82	88.21	−0.123	0.90	4.84	24.21	0.200	0.84	−151.22	64.56	−2.342	0.02
Age	−1.91	3.68	−0.519	0.61	−3.53	2.07	−1.704	0.10	−2.51	0.82	−3.039	0.003	−2.34	2.26	−1.033	0.31
Fat-Free Mass	0.03	0.01	2.953	0.01	0.01	0.01	1.618	0.10	0.02	0.002	9.501	<0.001	0.02	0.01	3.831	<0.001
**Model 3**	**Adjusted *R*^2^ = 0.79, *p* = 0.01**				**Adjusted *R*^2^ = 0.87, *p* < 0.001**				**Adjusted *R*^2^ = 0.71, *p* < 0.001**				**Adjusted *R*^2^ = 0.74, *p* < 0.001**			
	**Estimate**	**St Error**	**t Statistic**	***p* value **	**Estimate**	**St Error**	**t Statistic**	***p* value **	**Estimate**	**St Error**	**t Statistic**	***p* value **	**Estimate**	**St Error**	**t Statistic**	***p* value **
Constant	2354.36	1360.47	1.731	0.13	1335.92	2110.63	0.633	0.54	387.33	292.48	1.324	0.19	284.93	884.85	0.322	0.75
Height	−19.05	12.70	−1.500	0.18	−6.24	18.25	−0.342	0.74	2.02	2.38	0.848	0.40	−0.18	6.90	−0.027	0.98
Weight	1.12	8.00	0.140	0.89	4.74	14.54	0.326	0.75	1.24	1.49	0.837	0.40	6.62	3.21	2.062	0.05
Sex	−167.42	114.91	−1.457	0.19	−153.94	147.25	−1.045	0.31	1.62	24.22	0.067	0.95	−131.12	69.25	−1.893	0.06
Age	−5.45	4.00	−1.363	0.22	−5.71	6.94	−0.823	0.42	−2.61	1.01	−2.587	0.01	−2.52	2.63	−0.959	0.34
Fat-Free Mass	0.06	0.02	2.410	0.05	0.02	0.01	1.715	0.11	0.02	0.002	9.210	<0.001	0.02	0.01	3.518	<0.001
TG/HDL ratio	173.48	91.09	1.904	0.09	86.18	71.10	2.212	0.02	21.68	5.77	3.759	<0.001	30.22	14.62	2.068	0.04
**Model 4**	**Adjusted *R*^2^ = 0.75, *p* < 0.001**				**Adjusted *R*^2^ = 0.84, *p* = 0.01**				**Adjusted *R*^2^ = 0.73, *p* < 0.001**				**Adjusted *R*^2^ = 0.75, *p* < 0.001**			
	**Estimate**	**St Error**	**t Statistic**	***p* value **	**Estimate**	**St Error**	**t Statistic**	***p* value **	**Estimate**	**St Error**	**t Statistic**	***p* value **	**Estimate**	**St Error**	**t Statistic**	***p* value **
Constant	1294.20	1473.54	0.878	0.40	2317.58	4406.44	0.526	0.62	219.82	323.78	0.679	0.49	1294.2	1473.54	0.878	0.40
Height	−7.48	12.95	−0.577	0.57	−3.28	37.70	−0.087	0.93	3.68	2.6	1.418	0.16	−7.48	12.90	−0.577	0.57
Weight	5.85	10.37	0.564	0.58	−13.67	27.76	−0.493	0.64	0.49	1.67	0.292	0.77	5.85	10.37	0.564	0.58
Sex	−129.40	107.59	−1.203	0.25	−225.14	251.89	−0.894	0.41	−7.18	25.08	−0.286	0.78	−129.39	107.59	−1.203	0.25
Age	−1.33	5.13	−0.260	0.80	−10.47	12.08	−0.866	0.43	−2.76	1.42	−1.949	0.05	−1.33	5.13	−0.260	0.80
Fat-Free Mass	20.77	7.96	2.610	0.02	29.38	15.59	1.89	0.12	16.44	2.08	7.900	<0.001	20.77	7.96	2.610	0.02
TG/HDL ratio	67.78	27.58	2.457	0.03	162.37	114.72	1.42	0.22	23.41	6.49	3.61	<0.001	67.78	27.58	2.46	0.03
HOMA-IR	−2.46	5.52	−0.446	0.66	11.69	23.12	0.505	0.64	8.03	3.78	2.12	0.04	−2.64	5.52	−0.446	0.66

**Table 4 nutrients-14-05106-t004:** Characteristics of 645 Study Participants by Glycemic Status.

	Normoglycemic *n* = 384	Prediabetic *n* = 159	Diabetic *n* = 102	*p*-Value
Age (y)	36.9 ± 12.1	49.2 ±12.5	51.1 ± 9.6	<0.001
Weight (kg)	93.7 ± 17.9	102.7 ± 18.3	106.2 ± 17.9	<0.001
BMI (kg/m^2^)	32.6 ± 5.0	36.4 ± 5.6	36.2 ± 4.8	<0.001
Glucose (mg/dL)	87.9 ± 9.6	100.6 ± 15.7	143.9 ± 55.4	<0.001
Insulin (μIU/mL)	9.1 ± 7.0	9.6 ± 8.1	20.0 ± 15.3	<0.001
HOMA-IR (score)	2.0 ± 1.7	2.6 ± 2.3	7.9 ± 7.9	<0.001
Triglycerides (mg/dL)	90.1 ± 49.2	116.8 ± 54.9	162.2 ± 79.0	<0.001
HDL-cholesterol (mg/dL)	50.7 ± 13.5	47.2 ± 10.9	41.9 ± 12.1	<0.001
TG/HDL (ratio)	2.0 ± 1.5	2.7 ± 1.8	4.3 ± 2.7	<0.001
REE (kcal)	1717.3 ± 367.9	1701.1 ± 350.1	1879.8 ± 318.0	<0.001
% CHO (kcal)	32.9 ± 15.8	31.0 ± 14.9	26.7 ± 12.1	0.002
% Fat (kcal)	50.3 ± 16.2	48.4 ± 15.2	55.9 ± 13.2	0.001
Metabolic Equivalents (kcal/min)	0.77 ± 0.11	0.73 ± 0.18	0.74 ± 0.08	0.003
Respiratory Quotient (VCO_2_/VO_2_)	0.82 ± 0.06	0.82 ± 0.06	0.80 ± 0.05	0.068
Total Tissue Mass (kg) ^a^	96.3 ± 14.7	94.7 ± 13.8	104.5 ± 17.7	<0.001
Total Fat Mass (kg)	38.2 ± 10.9	44.7 ± 12.1	44.2 ± 10.7	<0.001
Total Fat (% tissue)	41.8 ± 9.6	45.3 ± 6.9	43.4 ± 6.5	<0.001
Trunk Fat (kg)	19.9 ± 7.1	25.9 ± 7.8	27.0 ± 6.8	<0.001
Trunk Fat (% tissue)	44.3 ± 10.3	49.7 ± 6.6	48.5 ± 5.6	<0.001
Android Fat (kg)	3.4 ± 1.4	4.6 ± 1.5	4.8 ± 1.4	<0.001
Android Fat (% tissue)	47.7 ± 10.9	53.2 ± 6.6	52.3 ± 5.9	<0.001
Visceral Adipose Tissue (kg)	1.3 ± 0.9	2.0 ± 1.1	2.4 ± 1.3	<0.001
Gynoid Fat (kg)	6.2 ± 2.1	7.2 ± 2.4	6.6 ± 2.1	<0.001
Gynoid Fat (% tissue)	44.1 ± 10.8	45.4 ± 8.7	42.6 ± 8.7	0.098
Total Lean Mass (kg)	53.6 ± 14.1	53.4 ± 10.8	57.3 ± 11.1	0.049
Trunk Lean Mass (kg)	25.2 ± 6.9	25.6 ± 5.2	27.8 ± 4.9	0.007
Android Lean Mass (kg)	3.7 ± 0.9	3.9 ± 0.8	4.3 ± 0.9	<0.001
Gynoid Lean Mass (kg)	8.5 ± 2.5	8.5 ± 1.8	8.7 ± 1.9	0.686

^a^ Total tissue mass by DXA = fat mass + lean mass + bone mineral content.

**Table 5 nutrients-14-05106-t005:** Characteristics of 645 Study Participants by Metabolic Syndrome Status.

Metabolic Syndrome *n* = 254		No Metabolic Syndrome *n* = 391	*p*-Value
Age (y)	46.8 ± 12.8	39.3 ± 13.1	<0.001
Weight (kg)	93.0 ± 17.3	105.4 ± 18.4	<0.001
BMI (kg/m^2^)	36.5 ± 4.9	32.5 ± 5.2	<0.001
Glucose (mg/dL)	111.4 ± 40.9	92.5 ± 20.3	<0.001
Insulin (μIU/mL)	16.9 ± 13.0	8.7 ± 7.0	<0.001
HOMA-IR (score)	5.5 ± 6.2	2.1 ± 2.4	<0.001
Triglycerides (mg/dL)	143.7 ± 70.9	81.0 ± 35.0	<0.001
HDL-cholesterol (mg/dL)	42.9 ± 10.8	52.7 ± 13.1	<0.001
TG/HDL (ratio)	3.7 ± 2.4	1.7 ± 0.9	<0.001
REE (kcal)	1813.0 ± 374.3	1689.9 ± 344.5	<0.001
% CHO (kcal)	30.6 ± 14.7	32.0 ± 15.5	0.267
% Fat (kcal)	50.0 ± 14.7	51.2 ± 16.3	0.374
Metabolic Equivalents (kcal/min)	0.74 ± 0.13	0.77 ± 0.12	0.029
Respiratory Quotient (VCO_2_/VO_2_)	0.82 ± 0.05	0.82 ± 0.06	0.828
Total Tissue Mass (kg) ^a^	104.9 ± 16.2	95.0 ± 14.5	0.938
Total Fat Mass (kg)	45.1 ± 10.8	37.9 ± 11.2	<0.001
Total Fat (% tissue)	44.4 ± 6.9	41.9 ± 9.5	<0.001
Trunk Fat (kg)	26.8 ± 7.2	20.0 ± 7.2	<0.001
Trunk Fat (% tissue)	49.5 ± 6.3	44.3 ± 9.9	<0.001
Visceral Adipose Tissue (kg)	2.1 ± 1.2	1.4 ± 1.0	<0.001
Android Fat (kg)	4.7 ± 1.4	3.4 ± 1.4	<0.001
Android Fat (% tissue)	52.9 ± 6.5	47.9 ± 10.8	<0.001*
Gynoid Fat (kg)	7.1 ± 2.1	6.1 ± 2.2	<0.001
Gynoid Fat (% tissue)	44.2 ± 8.5	44.2 ± 10.9	0.945
Total Lean Mass (kg)	56.6 ± 12.9	52.4 ± 12.5	<0.001
Trunk Lean Mass (kg)	26.7 ± 5.9	25.0 ± 6.1	0.004
Android Lean Mass (kg)	4.1 ± 0.9	3.7 ± 0.9	<0.001
Gynoid Lean Mass (kg)	8.8 ± 2.2	8.3 ± 2.2	0.010

^a^ Total tissue mass by DXA = fat mass + lean mass + bone mineral content.

**Table 6 nutrients-14-05106-t006:** Linear Regression Modeling in 645 Adults to Predict Resting Energy Expenditure by Glycemic Status.

Variables	Normoglycemia				Prediabetes				Diabetes Type 2			
**Model 1**	**Adjusted *R*^2^ = 0.65, *p* < 0.001**				**Adjusted *R*^2^ = 0.55, *p* < 0.001**				**Adjusted *R*^2^ = 0.62, *p* < 0.001**			
	**Estimate**	**St Error**	**t Statistic**	***p* value **	**Estimate**	**St Error**	**t Statistic**	***p* value **	**Estimate**	**St Error**	**t Statistic**	***p* value **
Constant	−1372.58	214.23	−6.407	<0.001	−613.78	420.35	−1.460	0.15	588.92	570.80	1.032	0.31
Age	−5.39	0.98	−5.521	<0.001	−3.69	1.59	−2.325	0.02	−8.50	2.24	−3.800	<0.001
Sex	62.24	25.82	2.410	0.02	68.98	53.08	1.299	0.20	174.89	67.12	2.606	0.01
Height	14.31	1.45	9.862	<0.001	8.52	2.97	2.867	0.01	2.49	4.06	0.613	0.54
Weight	8.21	0.78	10.548	<0.001	9.55	1.26	7.577	<0.001	9.90	1.54	6.449	<0.001
**Model 2**	**Adjusted *R*^2^ = 0.75, *p* < 0.001**				**Adjusted *R*^2^ = 0.60, *p* < 0.001**				**Adjusted *R*^2^ = 0.63, *p* < 0.001**			
	**Estimate**	**St Error**	**t Statistic**	***p* value **	**Estimate**	**St Error**	**t Statistic**	***p* value **	**Estimate**	**St Error**	**t Statistic**	***p* value **
Constant	205.00	276.14	0.742	0.46	787.46	495.81	1.588	0.12	852.83	594.70	1.434	0.16
Age	−2.67	0.94	−2.829	0.01	−3.39	1.54	−2.201	0.03	−6.86	2.38	−2.881	0.01
Sex	1.15	24.03	0.048	0.96	−64.98	57.92	−1.122	0.26	2.63	94.96	0.028	0.98
Height	2.24	2.04	1.098	0.27	−2.26	3.60	−0.628	0.53	0.19	4.24	0.045	0.96
Weight	3.79	0.95	3.975	<0.001	3.22	1.81	1.781	0.08	4.38	2.66	1.646	0.10
Fat-Free Mass	0.02	0.00	8.604	<0.001	0.02	0.01	4.869	<0.001	0.02	0.01	2.585	0.01
**Model 3**	**Adjusted *R*^2^ = 0.78, *p* < 0.001**				**Adjusted *R*^2^ = 0.66, *p* < 0.001**				**Adjusted *R*^2^ = 0.73, *p* < 0.001**			
	**Estimate**	**St Error**	**t Statistic**	***p* value **	**Estimate**	**St Error**	**t Statistic**	***p* value **	**Estimate**	**St Error**	**t Statistic**	***p* value **
Constant	453.32	299.07	1.516	0.13	1086.18	550.69	1.972	0.05	434.26	594.02	0.731	0.47
Age	−3.60	1.32	−2.729	0.01	−4.12	1.93	−2.137	0.04	0.28	2.78	0.099	0.92
Sex	2.85	23.98	0.119	0.91	−172.80	66.44	−2.601	0.01	−51.52	89.51	−0.576	0.57
Height	0.99	2.18	0.455	0.65	−4.76	4.00	−1.188	0.24	0.01	4.31	0.001	1.00
Weight	2.96	1.02	2.916	0.004	3.78	1.98	1.912	0.06	3.27	2.48	1.317	0.19
Fat-Free Mass	0.02	0.00	8.321	<0.001	0.03	0.01	5.220	<0.001	0.02	0.01	3.654	<0.001
TG/HDL ratio	31.57	7.50	4.212	<0.001	−5.79	11.98	−0.483	0.63	16.11	9.41	1.712	0.09
**Model 4**	**Adjusted *R*^2^ = 0.77, *p* < 0.001**				**Adjusted *R*^2^ = 0.67, *p* < 0.001**				**Adjusted *R*^2^ = 0.72, *p* < 0.001**			
	**Estimate**	**St Error**	**t Statistic**	***p* value **	**Estimate**	**St Error**	**t Statistic**	***p* value **	**Estimate**	**St Error**	**t Statistic**	***p* value **
Constant	179.75	342.41	0.525	0.60	224.97	1178.83	0.191	0.85	566.34	723.76	0.782	0.44
Age	−3.27	1.62	−2.021	0.04	−1.19	6.88	−0.173	0.87	−2.84	3.83	−0.743	0.46
Sex	−2.62	24.99	−0.105	0.92	−403.43	101.27	−3.984	0.001	−7.68	122.29	−0.063	0.95
Height	4.18	2.56	1.634	0.10	9.21	8.79	1.049	0.31	−0.11	5.14	−0.021	0.98
Weight	0.33	1.44	0.227	0.82	−2.53	7.88	−0.322	0.75	4.56	3.31	1.378	0.18
Fat-Free Mass	0.02	0.00	7.344	<0.001	0.02	0.01	2.056	0.05	0.02	0.01	2.624	0.01
TG/HDL ratio	25.57	8.12	3.150	0.002	22.57	8.61	2.980	0.004	12.61	5.84	2.374	0.01
HOMA-IR	18.70	7.83	2.388	0.02	−12.73	20.54	−0.620	0.55	−2.58	3.39	−0.761	0.45

**Table 7 nutrients-14-05106-t007:** Linear Regression Modeling in 645 Adults to Predict Resting Energy Expenditure by Metabolic Syndrome Status.

Variables	No Metabolic Syndrome				Metabolic Syndrome			
**Model 1**	**Adjusted *R*^2^ = 0.60, *p* < 0.001**				**Adjusted *R*^2^ = 0.66, *p* < 0.001**			
	**Estimate**	**St Error**	**t Statistic**	***p* value **	**Estimate**	**St Error**	**t Statistic**	***p* value **
Constant	−1563.30	215.27	−7.262	<0.001	−50.71	316.33	−0.16	0.87
Age	−3.90	0.90	−4.325	<0.001	−6.77	1.14	−5.912	<0.001
Sex	55.25	26.20	2.109	0.04	130.68	40.16	3.254	0.001
Height	15.70	1.45	10.796	<0.001	5.14	2.23	2.306	0.02
Weight	7.17	0.78	9.235	<0.001	10.78	1.02	10.564	<0.001
**Model 2**	**Adjusted *R*^2^ = 0.69, *p* < 0.001**				**Adjusted *R*^2^ = 0.70, *p* < 0.001**			
	**Estimate**	**St Error**	**t Statistic**	***p* value **	**Estimate**	**St Error**	**t Statistic**	***p* value **
Constant	−65.64	272.16	−0.241	0.81	1196.72	373.86	3.201	0.002
Age	−1.34	0.87	−1.545	0.12	−5.23	1.24	−4.210	<0.001
Sex	−12.02	24.34	−0.494	0.62	24.97	46.70	0.535	0.59
Height	4.21	2.00	2.103	0.04	−4.65	2.77	−1.682	0.09
Weight	2.81	0.92	3.052	0.00	6.09	1.37	4.443	<0.001
Fat-Free Mass	0.02	0.00	8.451	<0.001	0.02	0.00	5.471	<0.001
**Model 3**	**Adjusted *R*^2^ = 0.72, *p* < 0.001**				**Adjusted *R*^2^ = 0.75, *p* < 0.001**			
	**Estimate**	**St Error**	**t Statistic**	***p* value **	**Estimate**	**St Error**	**t Statistic**	***p* value **
Constant	202.97	315.17	0.644	0.52	1062.65	386.41	2.75	0.01
Age	−2.38	1.42	−1.680	0.09	−4.05	1.38	−2.945	0.004
Sex	−39.82	25.20	−1.580	0.12	35.77	47.01	0.761	0.45
Height	2.78	2.28	1.221	0.22	−4.34	2.85	−1.523	0.13
Weight	2.52	1.07	2.351	0.02	5.87	1.39	4.227	<0.001
Fat-Free Mass	0.02	0.00	7.883	<0.001	0.02	0.00	5.225	<0.001
TG/HDL ratio	31.35	13.16	2.382	0.02	16.59	6.66	2.492	0.01
**Model 4**	**Adjusted *R*^2^ = 0.70, *p* < 0.001**				**Adjusted *R*^2^ = 0.78, *p* < 0.001**			
	**Estimate**	**St Error**	**t Statistic**	***p* value **	**Estimate**	**St Error**	**t Statistic**	***p* value **
Constant	−87.01	348.05	−0.250	0.80	1082.45	514.49	2.104	0.04
Age	−1.46	1.57	−0.928	0.36	−2.26	2.22	−1.017	0.31
Sex	−37.23	25.73	−1.447	0.15	77.96	59.67	1.307	0.20
Height	6.27	2.59	2.419	0.02	−5.61	3.98	−1.408	0.16
Weight	−0.81	1.45	−0.562	0.58	8.32	2.91	2.86	0.006
Fat-Free Mass	15.95	2.17	7.34	<0.001	14.47	4.20	3.447	<0.001
TG/HDL ratio	24.86	13.78	1.855	0.05	19.28	8.04	2.399	0.01
HOMA-IR	9.45	4.59	2.061	0.04	−2.73	3.28	−0.830	0.41

## Data Availability

The data presented in this study may be available on request from the corresponding author. The data are not publicly shared due to privacy and ethical considerations based on the agreement as stated in the signed consent form.

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
