# Peer review of "TG/HDL Ratio Is an Independent Predictor for Estimating Resting Energy Expenditure in Adults with Normal Weight, Overweight, and Obesity"

_nutrients, 2022, doi:10.3390/nu14235106_

Round 1

Reviewer 1 Report

This study aimed to determine the contribution of insulin resistance biomarkers to the inter-individual variation in REE in per- 14 sons with varying body mass, and like a second objective, the author proposed to evaluate the contribution of insulin resistance biomarkers in the prediction of REE in persons of varying body mass when categorized by their glycemic and metabolic syndrome status. Baseline data from 645 adults.

Minor comments

Because sex is a variable that determines the distribution of adipose tissue differentially and therefore, the cardiometabolic risk, for example, the presence of insulin resistance, is different, the authors are recommended to present the data in table 1 stratified by sex.

Question: How do you explain the possible bias that they may have attributed to the fact that their groups stratified according to BMI, present significant differences in age and this tends to be higher as the BMI increases? Sustain in the discussion how they have corrected this bias attributed to age. As age increases, greater cardiovascular risk and alterations associated with metabolic syndrome and insulin resistance tend to appear

Author Response

We thank the reviewers for their thorough and thoughtful comments. We are confident the manuscript has addressed the reviewers' concerns. And, we believe the manuscript is improved as a result of responding to the reviewers' comments.  

We thank Reviewer 1 for these thoughtful comments.

Minor comments

Because sex is a variable that determines the distribution of adipose tissue differentially and therefore, the cardiometabolic risk, for example, the presence of insulin resistance, is different, the authors are recommended to present the data in table 1 stratified by sex.

As suggested by the reviewer, we have created a revised version of Table 1. However, because all our models have accounted for sex, which only predicted 3-4% of the total variance in REE, we do not feel that Table 1 should be replaced, but instead, we have provided this additional information for readers who would like to see the differences in variables by sex as a Supplemental Table.

Question: How do you explain the possible bias that they may have attributed to the fact that their groups stratified according to BMI, present significant differences in age and this tends to be higher as the BMI increases? Sustain in the discussion how they have corrected this bias attributed to age. As age increases, greater cardiovascular risk and alterations associated with metabolic syndrome and insulin resistance tend to appear.

We appreciate the reviewer’s concern. The potential bias attributed to the association between age and resting energy expenditure, the primary outcome, has been addressed by including age in all regression modeling, as presented in Tables 3, 6, 7. Of note, a study published in 2016 by Geisler et al PMID: 27258302 showed that the relationship between age and REE is a factor of decrease in fat free mass, which also was included in our regression models. Like our findings the variance in REE associated with age was quite small, only 2%   

Reviewer 2 Report

The authors aimed to determine whether TG:HDL and HOMA were independent predictors of REE in a heterogenous sample of people with varying degrees of obesity, and glycemic control status. The manuscript has potential but I think the authors focus on TG:HDL being a surrogate of insulin resistance is a red herring as TG:HDL is more of an indicator of the capacity of the liver to use and repackage lipids than insulin resistance (although they are obviously related). To illustrate this point, I suggest the authors reverse the order of the variables in regression model with TG:HDL being inserted in the block prior to HOMA. I predict that HOMA-IR would not add to the model beyond TG:HDL. In this light, lipid handling dysfunction in the liver needs to feature more prominently throughout the manuscript and the potential modifying effects of cardiorespiratory fitness should then also be mentioned with the references provided below. Finally, please adjust the presentation of tables as some are very difficult to follow.

I have specific comments below.

Line 11. Should read large body mass or high body mass index. 

To determine a, you need to do a partial correlation analysis between REE and insulin resistance controlling for BMI.

Line 43. Remove the “most” from “most likely”. 

Line 46 to 48. From how much to how much?

Line 48 to 49: please remove reference to Pima Indians and replace with native Americans or Pima peoples. 

Line 51 to 53. Please indicate direction of association.

Line 46 to 55. I think this paragraph needs a rewrite. There are conflicting data on whether insulin resistant persons have higher REE than insulin sensitive persons (For PMID: 17092386 against 25312871). I think highlighting conlficting data will help add clinical relevance to your study as you, with such a large sample, may be able to determine whether there indeed is an association or difference in REE across different BMIs. Especially since you don’t have T2D participants. 

Line 56 to 57, I think this paragraph also needs a rewrite. Here the authors assume that the relationship between TG/HDL ratio is a suitable surrogate for insulin resistance and consequently REE but TG/HDL may actually be more closely related to fatty liver disease (PMID: 30711017), which is also bidirectional related with insulin resistance. In fact adults with NAFLD have been shown to have higher REE than those without after controlling for Age Sex, FM, and FFM (PMID: 31592493). I think the paper rationale can be strengthened by focusing on biomarkers which predict REE rather than trying to draw an association between lipid levels and insulin resistance and then to REE. The lipid angle alone is interesting enough. 

Line 114 to 116. How were these participants selected? How can there be < 1.5 % difference for total fat, lmm, etc for people with varying degrees of obesity?

Table 1. The table needs to be adapted to landscape as it is very difficult to see which values line up with the relevant variables. How did you determine %CHO,PRO,FAT? What value is RQ reported in? Why are there p values for BMI when the table is stratified by BMI? 

Table 3. Very difficult to follow. Please change to landscape. 

Results, to me, the biggest take away here is the impaired lipid metabolism manifested by TG:HDL ratio contributing to higher REE rather than HOMA to REE. In my opinion, the data do not support the notion that insulin resistance contributes to REE as in most models, the addition of HOMA has minimal effect of adjusted R2. 

Discussion. Both HOMA and TG:HDL do not independently predict REE to the same degree. HOMA, at most, improves models by 2% and in some instances decreases the models prediction. Whereas TG:HDL improves models, particularly when stratified by glycemic status. Thus the opening paragraph of the discussion needs to be amended to highlight that the relative contribution of HOMA and TG:HDL are different with the latter being a greater predictor. 

I think the authors should at least acknowledge in the discussion, that variance in cardiorespiratory fitness may have contributed to differences in the results. Cardiorespiratory fitness is inversely associated with fatty liver disease (hepatic manifestation of metabolic syndrome)(PMID: 33858477) which itself is independently predicts TG:HDL (PMID: 30711017). Thus, those with poor CRF would have have impaired lipid handling capacity (manifested in abnormal TG:HDL) and higher REE. 

Author Response

We thank Reviewer 2 for the thorough and thoughtful comments. We are grateful that the manuscript has been much improved by the revisions resulting from these comments. 

Reviewer 2

The manuscript has potential but I think the authors focus on TG:HDL being a surrogate of insulin resistance is a red herring as TG:HDL is more of an indicator of the capacity of the liver to use and repackage lipids than insulin resistance (although they are obviously related). To illustrate this point, I suggest the authors reverse the order of the variables in regression model with TG:HDL being inserted in the block prior to HOMA. I predict that HOMA-IR would not add to the model beyond TG:HDL.

As suggested by the reviewer, we have reversed the order of the variables in the regression modeling, and now report that HOMA-IR did not add to the model, please see Results lines 188-190 and 205-206, and revised Tables 3,6,7.

In this light, lipid handling dysfunction in the liver needs to feature more prominently throughout the manuscript and the potential modifying effects of cardiorespiratory fitness should then also be mentioned with the references provided below.

As suggested by the reviewer, we have now featured TG/HDL and the issue of lipid handling dysfunction by revising the title, the intro text, the regression modeling, the results, and the discussion sections. Most references provided throughout this reviewers’ comments have been incorporated.

The concern raised regarding cardiorespiratory fitness has been addressed as a limitation in the discussion section since this was not a measured variable in this study, please see lines 283-287.

Finally, please adjust the presentation of tables as some are very difficult to follow.

Yes, they tables will be landscape in the published article, but we have provided them here as pictures so they can fit in the upload of the word document. When we print out the tables, all the data is present. 

I have specific comments below.

Line 11. Should read large body mass or high body mass index. 

As suggested by the reviewer, term changed to high body mass index, now line 3.

To determine a, you need to do a partial correlation analysis between REE and insulin resistance controlling for BMI.

As suggested by the reviewer, we have reworded ”a” to clarify the methodology, now line 6-7.

Line 43. Remove the “most” from “most likely”. 

As suggested by the reviewer, “most” removed, now line 33.

Line 46 to 48. From how much to how much?

As suggested by the reviewer, how much (by 3%) has been added, now line 38.

Line 48 to 49: please remove reference to Pima Indians and replace with native Americans or Pima peoples. 

As suggested by the reviewer, reworded to Pima peoples, now line 41.

Line 51 to 53. Please indicate direction of association.

As suggested by the reviewer, we added positive to indicate direction of association, now line 44.

Line 46 to 55. I think this paragraph needs a rewrite. There are conflicting data on whether insulin resistant persons have higher REE than insulin sensitive persons (For PMID: 17092386 against 25312871). I think highlighting conflicting data will help add clinical relevance to your study as you, with such a large sample, may be able to determine whether there indeed is an association or difference in REE across different BMIs. Especially since you don’t have T2D participants. 

 As suggested by the reviewer, we have rewritten this paragraph, please see lines 37-39

Line 56 to 57, I think this paragraph also needs a rewrite. Here the authors assume that the relationship between TG/HDL ratio is a suitable surrogate for insulin resistance and consequently REE but TG/HDL may actually be more closely related to fatty liver disease (PMID: 30711017), which is also bidirectional related with insulin resistance. In fact adults with NAFLD have been shown to have higher REE than those without after controlling for Age Sex, FM, and FFM (PMID: 31592493). I think the paper rationale can be strengthened by focusing on biomarkers which predict REE rather than trying to draw an association between lipid levels and insulin resistance and then to REE. The lipid angle alone is interesting enough.

 As suggested by the reviewer, we have rewritten this paragraph, and added an additional paragraph, lines 63-73.

Line 114 to 116. How were these participants selected? How can there be < 1.5 % difference for total fat, lmm, etc for people with varying degrees of obesity?

We appreciate the reviewers concern. The CV was calculated for one variable (each DXA measurement) upon repeated measures. For these 12, each person was measured, repeatedly, under the same conditions (time of day, upon voiding, same hours NPO, in hospital gowns, etc.) Each individual is compared to themselves for each DXA measure, which has been repeated, not compared to the others in the group of 12. The 12 were selected randomly from the group of individuals who have a range of obesity (BMI ≥ 30). While it would be great to perform repeated measures for all members of the group, that was not feasible, with regard to time and cost, and is not the specific aim of the study. The coefficient of variation between the repeated measures, then averaged for the group, reflects the precision and repeatability of our DXA measurements for persons with obesity who are from the study population. It’s reported as a quality control indicator. We have added text, please see line 115.  

Table 1. The table needs to be adapted to landscape as it is very difficult to see which values line up with the relevant variables.

Yes, the table is landscape, it just is not showing up that way in the pdf, but the layout in the journal of the published manuscript will show all variables and all data. When we print out the tables all the data is present.    

How did you determine %CHO,PRO,FAT?

The calorimeter (Parvo metabolic cart) first quantifies VCO2 expired and VO2 inspired. REE is determined by the Weir equation. Substrate oxidation rates, CHO and FAT values, are automatically calculated using the Frayn methods. Each macronutrient has a specific energy released per liter of O2 consumed and oxidation rates are calculated from the respiratory exchange measurements. Protein oxidation is calculated indirectly, based on a complete 24-hour urine collection and lab determination of nitrogen excretion. This value can then be entered into the software to enable the final CHO and FAT values (percentage of energy expenditure) being adjusted for PRO. Please see revised lines 128-131.

What value is RQ reported in?

RQ is a ratio of VCO2 production to VO2 consumption, please see Tables for measurement unit.  

Why are there p values for BMI when the table is stratified by BMI? 

We wish to inform the reader of the average BMI and distribution of BMI in each group, and to confirm for the reader that the groups are significantly different by BMI category based on the data.  

Table 3. Very difficult to follow. Please change to landscape. 

Yes, the table is landscape, it just is not showing up that way in the pdf, but the layout in the journal of the published manuscript will show all variables and all data. When we print out the tables all the data is present.      

Results, to me, the biggest take away here is the impaired lipid metabolism manifested by TG:HDL ratio contributing to higher REE rather than HOMA to REE. In my opinion, the data do not support the notion that insulin resistance contributes to REE as in most models, the addition of HOMA has minimal effect of adjusted R2. 

We have revised the results based on this reviewer’s suggestions above, see revised Results lines 188-190 and 205-206

Discussion. Both HOMA and TG:HDL do not independently predict REE to the same degree. HOMA, at most, improves models by 2% and in some instances decreases the models prediction. Whereas TG:HDL improves models, particularly when stratified by glycemic status. Thus the opening paragraph of the discussion needs to be amended to highlight that the relative contribution of HOMA and TG:HDL are different with the latter being a greater predictor. 

As suggested by the reviewer, we have rewritten this sentence, now line 218

 I think the authors should at least acknowledge in the discussion, that variance in cardiorespiratory fitness may have contributed to differences in the results. Cardiorespiratory fitness is inversely associated with fatty liver disease (hepatic manifestation of metabolic syndrome)(PMID: 33858477) which itself is independently predicts TG:HDL (PMID: 30711017). Thus, those with poor CRF would have impaired lipid handling capacity (manifested in abnormal TG:HDL) and higher REE. 

We agree with the reviewer that cardiorespiratory fitness, as well as physical activity level, may be a factor affecting REE. Since we have not directly measured these, we do acknowledge this possibility in the limitations section, please see line 284-288

Reviewer 3 Report

Nutrients 1995983 HOMA-IR vs TG/HDL, IR & REE

Abstract: Consider editing ‘when categorized by their glycemic and metabolic syndrome status’ to what may actually being measured, systemic redox imbalance and redox balance. If you can reference why TG may be a measure of dietary highly processed Maillard end-product (MEP or AGE, ALE) containing food/beverage-induced systemic oxidative stress, and HDL may be a measure of dietary unheated to minimally processed MEP containing food/beverage-induced systemic reductive stress.  

Introduction: First paragraph, may reflect editing ‘it is most likely that a significant portion of the unexplained inter-individual variance in REE is related to cardiometabolic risk factors, such as insulin resistance’ and a yet to be uncovered possibly more precise indicator that is readily accessible retroactively from decades old chart reviews.

Abstract: Consider editing ‘and cardio metabolic biomarkers’ to and a recently discovered measure of diet-induced systemic oxidative stress.

Abstract: Consider editing ‘however, TG/HDL ratio was a more robust predictor’ of the magnitude of oxidative stress and insulin resistance, and the dietary and associated management of many top causes of morbidity and mortality globally.

Introduction: Last paragraph, consider editing ‘when categorized by their glycemic and metabolic syndrome, and their main activators, diet induced systemic oxidative stress.

Discussion: Consider editing accordingly.

This paper has the potential of replacing the highly clinically obscure term metabolic syndrome with the precise clinical managing factor of many of the leading causes of sickness and death worldwide, diet-induced systemic oxidative stress.

Author Response

We thank the reviewers for their thorough and thoughtful comments. We are confident the manuscript has addressed the reviewers' concerns. And, we believe the manuscript is improved as a result of responding to the reviewers' comments.  

Reviewer 3

Abstract: Consider editing ‘when categorized by their glycemic and metabolic syndrome status’ to what may actually being measured, systemic redox imbalance and redox balance. If you can reference why TG may be a measure of dietary highly processed Maillard end-product (MEP or AGE, ALE) containing food/beverage-induced systemic oxidative stress, and HDL may be a measure of dietary unheated to minimally processed MEP containing food/beverage-induced systemic reductive stress. 

We appreciate the reviewer’s perspective and concern for accurate information. We agree that the clinical criteria used to identify metabolic syndrome does not account for dietary factors and that there remains scientific debate about the cause of the metabolic syndrome and what it represents. However, we aim to be transparent regarding how participants were categorized – which was based on the current definition of metabolic syndrome as stated by the WHO, NIH NHLBI, the AHA, Diabetes Canada Clinical Practice Guidelines, US NCEP, International Diabetes Federation.  

Introduction: First paragraph, may reflect editing ‘it is most likely that a significant portion of the unexplained inter-individual variance in REE is related to cardiometabolic risk factors, such as insulin resistance’ and a yet to be uncovered possibly more precise indicator that is readily accessible retroactively from decades old chart reviews.

As suggested by the reviewer, we have edited this sentence in the first paragraph in the Introduction, now line 34-36

Abstract: Consider editing ‘and cardio metabolic biomarkers’ to and a recently discovered measure of diet-induced systemic oxidative stress.

We appreciate the reviewer’s comments. It is important to recognize that we did not measure systemic redox balance/imbalance or any form of oxidative stress in this study.      

Abstract: Consider editing ‘however, TG/HDL ratio was a more robust predictor’ of the magnitude of oxidative stress and insulin resistance, and the dietary and associated management of many top causes of morbidity and mortality globally.

We appreciate the reviewer’s suggestion. To be clear, this study was not designed to investigate the impact of dietary factors on resting energy expenditure. Most evidence shows that the composition of the diet is not a significant predictor of resting energy expenditure (e.g. PMID: 22627912, 15598683, 28678735). Instead, diet is a component of total energy expenditure, due to the contribution of diet induced thermogenesis, which comprises ~10% of total energy expenditure.

 Discussion: Consider editing accordingly.

Since we have not measured oxidative stress, nor have we measured mitochondrial dysfunction, we have limited the extent to which this is mentioned in the Discussion. However, we have expanded the Discussion content with regard to TG/HDL as not just a biomarker of insulin resistance but impaired lipid metabolism and fatty liver disease that interfere with insulin sensitivity, and linked liver lipid dysfunction to the development of oxidative stress, please see lines 259-264 and lines 281-283.      

This paper has the potential of replacing the highly clinically obscure term metabolic syndrome with the precise clinical managing factor of many of the leading causes of sickness and death worldwide, diet-induced systemic oxidative stress.

We appreciate the reviewers’ perspective regarding diet-induced oxidative stress. We agree that the typical “western” diet is not balanced, and is a driver of morbidities, with regard to dietary saturated fat and simple carbohydrate intake. We also agree that both diet and physical activity can affect the risk factors that cluster together as the metabolic syndrome. And, we acknowledge that metabolic syndrome definitions do not consider dietary factors.

However, with regard to replacing the term metabolic syndrome, which is used in the Title of over 27,000 manuscripts available in PubMed, we have not acquired the data, nor have we performed the analysis, that would justify using diet-induced systemic oxidative stress as a replacement for metabolic syndrome.

To our knowledge, the phenomenon of diet-induced oxidative stress has been investigated primarily in preclinical models when there is exposure specifically to what is called a high fat diet, and is really a high saturated fat diet, most often from adding a source of palmitate. In fact, a PubMed search using (diet-induced oxidative stress [Title]) yielded 34 publications, all except 3 report findings from high fat diet induced oxidative stress. In contrast, the respiratory quotient of our participants reflects a more typical mixed diet, not the experimental high fat diets used in these oxidative stress studies.   

Round 2

Reviewer 2 Report

Please highlight in the abstract that the relationship between REE and TG/HDL ratio is independent of fat-free mass.

Table 3 should have HOMA-IR written as HOMA-IR by iteself or HOMA-IR score, but not HOMA-IR Ratio.

Author Response

Abstract text revised as suggested, and Table 3 corrected as suggested.